# Biological Evaluation of 4-(1H-triazol-1-yl)benzoic Acid Hybrids as Antioxidant Agents: In Vitro Screening and DFT Study

**Hatem A. Abuelizz** [1], **Hanan A. A. Taie** [2], **Ahmed H. Bakheit** [1], **Mohamed Marzouk** [3], **Mohamed M. Abdellatif** [4] and **Rashad Al-Salahi** [1,*]

1   Department of Pharmaceutical Chemistry, College of Pharmacy, King Saud University, P.O. Box 2457, Riyadh 11451, Saudi Arabia; Habuelizz@ksu.edu.sa (H.A.A.); abujazz76@gmail.com (A.H.B.)

2   Plant Biochemistry Department, Agricultural and Biology Research Institute, National Research Centre, 33 El-Bohouth St. (Former El-Tahrir St.), Dokki, Cairo 12622, Egypt; hanan_taie@yahoo.com

3   Chemistry of Tanning Materials and Leather Technology Department, Chemical Industries Research Institute, National Research Centre, 33 El-Bohouth St. (Former El-Tahrir St.), Dokki, Cairo 12622, Egypt; msmarzouk@yahoo.co.uk

4   Department of Chemistry, Graduate School of Science, Tokyo Metropolitan University, 1-1 Minami Osawa, Tokyo 192-0397, Japan; Mohamed-soliman@tmu.ac.jp

*   Correspondence: ralsalahi@ksu.edu.sa; Tel.: +966-114677194

**Abstract:** Fourteen triazole benzoic acid hybrids were previously characterized. This work aimed to screen their in vitro antioxidant activity using different assays, i.e., DPPH (1,1-diphenyl-1-picrylhydrazyl), reducing the power capability, FRAP (ferric reducing antioxidants power) and ABTS (2,2′-azino-bis(3-ethylben zothiazoline-6-sulfonate) radical scavenging. The 14 compounds showed antioxidant properties in relation to standard BHA (butylated hydroxylanisole) and Trolox (6-hydroxy-2,5,7,8-tetramethylchroman-2-carboxylic acid). Higher antioxidant activity was observed by the parent (**1**) at a concentration of 100 μg/mL ($89.95 \pm 0.34$ and $88.59 \pm 0.13\%$) when tested by DPPH and ABTS methods in relation to BHA at 100 μg/mL ($95.02 \pm 0.74$ and $96.18 \pm 0.33\%$). The parent (**2**) demonstrated remarkable scavenging activity when tested by ABTS ($62.00 \pm 0.24\%$), however, **3** was less active ($29.98 \pm 0.13\%$). Compounds **5**, **6**, **9**, and **11** exhibited good scavenging activity compared to **1**. DFT studies were performed using the B3LYP/6-311++g (2d,2p) level of theory to evaluate different antioxidant descriptors for the targets. Three antioxidant mechanisms, i.e., hydrogen atom transfer (HAT), sequential electron transfer proton transfer (SETPT) and sequential proton loss electron transfer (SPLET) were suggested to describe the antioxidant properties of **1–14**. Out of the 14 triazole benzoic acid hybrids, **5**, **9**, **6**, and **11** showed some good theoretical results, which were in agreement with some experimental outcomes. Based on the computed (PA and ETE) and (BDE and IP) values in (SPLET) and (HAT and SETPT) mechanisms, respectively, compound **9** emerged has having good antioxidant activity.

**Keywords:** triazole derivatives; DPPH; reducing power; FRAP; ABTS; DFT and SPLET mechanism

## 1. Introduction

Most drugs and biologically relevant compounds possess triazole nucleus privileged precise in their pharmacological purposes [1–3]. Triazole is a biologically imperious platform known to be related with significant pharmacological activities such as antidepressant [4], anticonvulsant [5,6], antioxidant [7,8], antimicrobial [9,10], antiviral [11–13], analgesic [14], anti-inflammatory [14,15], hypoglycemic [16], anticancer [17,18], antihistamine [19,20], pesticidal-insecticidal [21,22], CNS depressant [23], and antihypertensive properties [24,25]. The significant therapeutic values of triazoles combined with their efficient synthetic procedures made them very interesting units for biological investigation. Thus, medicinal chemists have incorporated several active heterocyclic platforms as

quinazoline, pyridine, pyrimidine, thiophene, and benzene into the triazole entity using developing synthetic approaches to elaborate a library of triazole derivatives with different bioactivities. A literature survey revealed that numerous available clinical drugs, such as antifungal agents (fluconazole, posaconazole itraconazole, voriconazole, and ravuconazole) and antiviral (ribavirin), anti-migraine and headache (rizatriptan), antianxiety disorders (alprazolam) and tranquilizer (estazolam) originated from a triazole entity [26]. Triazole structure-activity relationship studies identified the correlation of their specific structural features (dipole moment character, H-bonding capability, rigidity, and in vivo/vitro stability conditions) with their pharmacological targets. It was reported that triazole-bridged benzoheterazole dendrimers with a bisphenol/benzophenone core unit have shown antioxidant properties in relation to those of ascorbic acid as a reference drug when evaluated by ABTS and DPPH assays [27]. Hybridization of 1,2,3-triazole pharmacophoric moiety with hydrazide, carbonitrile, and hydrazone units was positively influenced on free radical scavenging activity, and showed the best anticancer effects [28,29]. The Schiff bases of 4-aminotriazole derivatives demonstrated potential free radical scavenger effects, whereas the N-substituted triazole attached indole/chalcone hybrids and 1,2,4-triazole-3-thiones reported to exhibit significant antioxidant activity on DPPH radicals and were found to be potent anticancer agents [30,31]. Incorporation of 1,2,4-triazole ring with quinazoline and pyridopyrimidine units was proved to be favorable for the antioxidant and cytotoxicity activities [32–34].

Throughout our research, we have noticed that the 1,2,4-triazole or 1,2,3-triazole derivatives showed antioxidant properties [28–35] and demonstrated cytotoxic activity as well. In a previous work, the molecular hybridization technique was employed to incorporate the triazole unit with isothiocyanate and substituted benzylidene groups into a single molecule; the resulting targets have the potential to exhibit potent antiproliferative activity against HCT-116 and MCF-7 human cancer cell lines and showed very weak cytotoxicity against normal cells [35]. Given the above facts, and continuing our long-term interest in biologically significant 1,2,4-triazole derivatives (Figure 1), we report herein the antioxidant activity of 4-(1*H*-triazol-1-yl)benzoic acid hybrids (**1–14**) using several assays as DPPH, reducing power capability, ferric reducing power, and ABTS radical scavenging. Density function theory (DFT) calculations were carried out based on set B3LYP/6-311++g (2d,2p) level of theory to obtain numerous molecular properties of the target compounds. To explain the structure-antioxidant relationship of the targets, various antioxidant descriptors such as ionization potential (IP), electron affinity (EA), hardness (η), softness (S), electrophilicity (ω), electronegativity (χ), and chemical potential (μ) were calculated. In addition, the bond dissociation enthalpy (BDE), proton dissociation enthalpy (PDE), PA (proton affinity), and ETE (electron-transfer enthalpy) parameters for targets **1–14** were determined to elucidate their antioxidant mechanisms. The highest occupied and the lowest unoccupied molecular orbitals (HOMO and LUMO) energies were determined, as well.

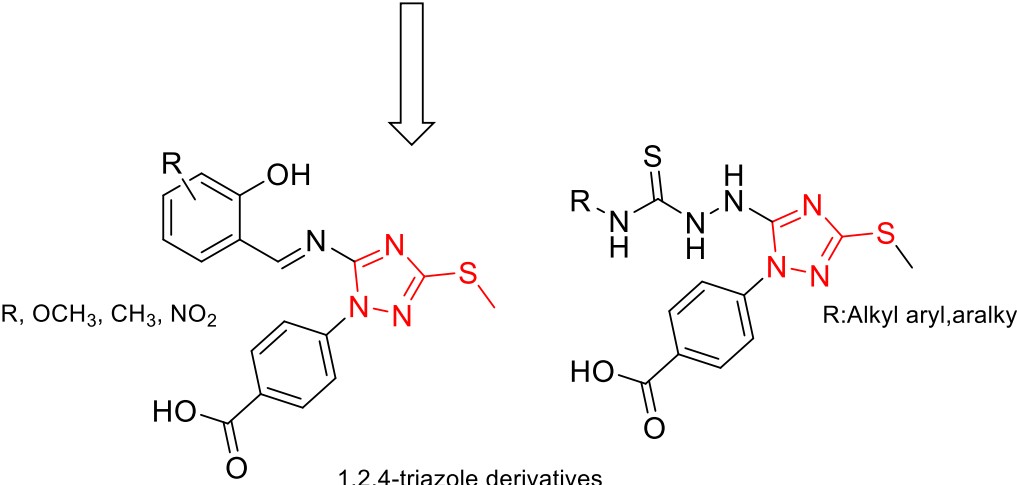

**The reported antioxidant triazole derivatives**

**The designed triazole derivatives**

**Figure 1.** The reported and designed triazole derivatives.

## 2. Materials and Methods

### 2.1. Antioxidant Activity Investigation

The detailed procedures for performing DPPH, reducing power capability, FRAP, and ABTS radical scavenging assays were described in our previous work [36–39].

### 2.2. Computation of Antioxidant Descriptors

The ideal mechanism of free radical scavenging for the target molecules **1–14** was determined by computing the antioxidant descriptors as follows:

The BDE was computed with Equation (1) under normal parameters, standard atmospheric pressure and 298.15 K. For instance, when a chemical bond is broken under standard conditions, BDE indicates the standard reaction enthalpy change at a specific temperature [40]. The O–H bond is less stable at lower BDE values [41].

$$BDE = H_{radical} + H_{electron} - H_{neutral}. \tag{1}$$

The adiabatic ionization potential (AIP) was calculated in Equation (2). The capability of the antioxidant compound is inversely proportional to the AIP value.

$$AIP = H_{cation\ radical} + H_{electron} - H_{neutral}. \tag{2}$$

The PDE was determined according to Equation (3) and compounds with lower PDE values are more exposed to proton abstraction [42].

$$PDE = H_{radical} + H_{H^+} - H_{cation\ radical}. \tag{3}$$

The PA and ETE were calculated by Equations (4) and (5).

$$PA = H_{anion} + H_{H^+} - H_{neutral}, \tag{4}$$

$$BDE = H_{radical} + H_{electron} - H_{anion}, \tag{5}$$

where $H_{radical}$ is the phenoxy radical total enthalpy, $H_H$ is the hydrogen atom total enthalpy, $H_{neutral}$ is the neutral compound enthalpy, $H_{H^+}$ is the proton total enthalpy, $H_{cation\ radical}$ is the cation radical total enthalpy, $H_{electron}$ is the electron total enthalpy, $H_{anion}$ is total enthalpy of the anion.

The total species enthalpies were calculated as the sum of total electronic energy, zero-point energy and the translational, rotational, and vibrational contributions to the total enthalpy as presented in Equation (6). In order to convert the energy to enthalpy, the RT (PV-work) term was added [43].

$$H = E_0 + ZPE + H_{trans} + H_{rot} + H_{vib} + RT. \tag{6}$$

The translational, rotational, and vibrational contributions to enthalpy are $H_{trans}$, $H_{rot}$, and $H_{vib}$. $E_0$ represents the total energy at zero K, whereas ZPE represents the zero-point vibrational energy. When computing the antioxidant descriptors listed above, the following values were utilized to carry out the calculations: In the case of $H(H^{\bullet})_{vacuum}$, the value is $-1312.479673$ kcal.mol$^{-1}$; in the case of $H(H^+)_{vacuum}$, the value is $6.1961805$ kcal.mol$^{-1}$; in the case of $H(e^-)_{vacuum}$, the value is $3.1454$ kcal.mol$^{-1}$; in the case of $H(e^-)_{vacuum}$, the value is $3.1454$ kcal.mol$^{-1}$; in the case of $H(H^{\bullet})_{water}$, the value is $-3.9908$ kJ/mol; in the case of $H(H^+)_{water}$ the value is $-1090.0027$ kcal.mol$^{-1}$; in the case of $H(e^-)_{hydr}$, the value is $-105$ kcal.mol$^{-1}$ [44–47].

The geometry optimization was done on all molecular structures in the gas phase at the DFT/B3LYP/6-311$^{++}$G(d,p) level of theoretical. This study used methanol as the physiological medium in vitro and the self-consistent reaction field approach using a polarized continuum model (PCM) [47]. According to the reported literature [48] all

chemical descriptors such as softness hardness, chemical potential, and electrophilicity were calculated. Using the Gaussian 09 program package, all calculations were carried out.

### 3. Results and Discussion

Triazole benzoic acid hybrids (**1–14**) were described in our previous work as predicted in Scheme 1 and Table 1. Parents **1** and **2** resulted from the reaction of *N*-cyanoimido(dithio)carbonates with 4-hydrazinobenzoic acid. Oxidation of the thiomethyl moiety in **1** into methylsulfonyl (**3**) was successfully achieved using hydrogen peroxide. The hydrazone derivatives **4–12** were obtained by the reaction of **1** and **2** with several aldehydes, whereas the products **13** and **14** afforded smoothly upon treatment of **1** with benzyl(phenethyl)isothiocyanate [35].

*3.1. DPPH Radical Scavenging Activity*

The triazole derivatives **1–14** were examined for their free scavenging properties by using the DPPH method. This assay was carried out to test their antioxidant abilities and determine their behaviors as proton radical scavengers or hydrogen suppliers. When the DPPH combines with an antioxidant substance (reducing agent), the electron pairing goes off, and the disappearance of solution color is confirmed by a decrease in absorbance (517 nm). Results in Figure 2 illustrated the DPPH radical scavenging activity of 4-(1*H*-triazol-1-yl)benzoic acid hybrids **1–14** with BHA as standard antioxidant for comparison purpose. The main compounds **1–3** showed clear variation in their DPPH radical scavenging activity, where the thioether derivative **1** exhibited the highest radical scavenging effect (89.95 $\pm$ 0.343%) with IC$_{50}$ of 55.59 μg/mL, in relation to BHA (95.02 $\pm$ 0.74%) at the same concentration. Compound **2** came second in its radical scavenging activity scoring (31.47 $\pm$ 0.165%), while sulfonyl derivative **3** recorded the lowest DPPH radical scavenging activity (16.47 $\pm$ 0.161%) at the concentration of 100 μg/mL.

i: aldehydes, EtOH, reflux  ii: isothiocyanates, EtOH, reflux

**Scheme 1.** The synthetic routes for the 4-(1*H*-triazol-1-yl)benzoic acid hybrids **1–14**.

**Table 1.** Chemical structures of compounds **1**–**14**.

(**1**)

(**2**)

(**3**)

(**4**)

(**5**)

(**6**)

(**7**)

(**8**)

(**9**)

(**10**)

(**11**)

(**12**)

(**13**)

(**14**)

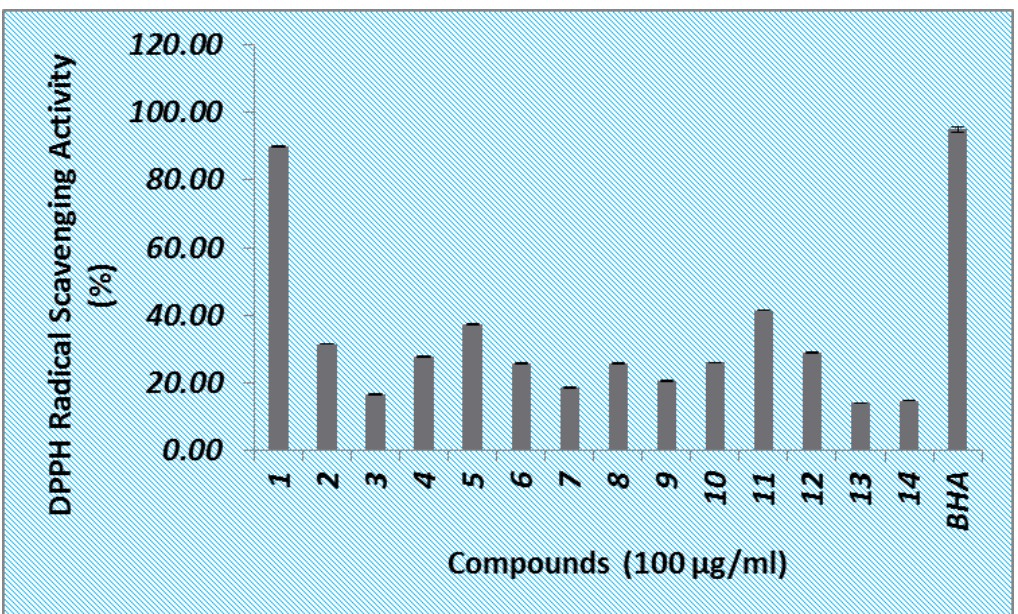

**Figure 2.** DPPH radical scavenging activity of **1–14**. Data are means ± standard deviation of triplicate experiments.

Generally, substitutions in compound **1** resulted in a remarkable decrease in the radical scavenging activity for all thioether derivatives (**4–9**, **13**, and **14**). Among them, the 2-hydroxy-5-methoxy product (**5**) exhibited good activity (37.18 ± 0.110%), followed by compounds **4**, **6**, **8**, and **9** that demonstrated 27.51 ± 0.182, 25.77 ± 0.11, 25.61 ± 0.17, and 20.49 ± 0.16%, respectively. The lowest radical scavenging activity recorded by compounds **7**, **13**, and **14** (18.54 ± 0.098, 14.01 ± 0.086 and 14.63 ± 0.075%, respectively). Concerning the substitutions of compound **2**, an increase in the radical scavenging activity was observed by **11** (41.40 ± 0.11%), while compounds **10** and **12** showed a slight decrease in the radical scavenging activity compared to their parent **2** at the same concentration.

*3.2. Reducing Power Ability*

Reducing power ability is one of the most important assays for determination of the antioxidant activity of compounds. The parent thioether **1** revealed the highest reducing power capability in comparison with the other two parents (**2** and **3**) and relative to all investigated products (Table 2). It recorded an activity of 1.43 ± 0.007, comparable to that of the standard BHA (1.76 ± 0.062) at the same concentration (100 µg/mL). From Table 2, compounds **2** and **3** exhibited lower reducing ability (0.270 ± 0.004 and 0.31 ± 0.007) than **1**. Chemical transformations on the structure of **1** produced **6** and **9** with good reducing activity (0.77 ± 0.014 and 0.58 ± 0.013) in relation to other products, however, lower than the parent **1**. The 2-hydroxy-5-bromo derivative (**7**) recorded the lowest scavenging ability of 0.16 ± 0.012. On the other hand, by substitution on compound **2**, a slight increase in the reducing activity (0.272 ± 0.012 and 0.310 ± 0.010) was recorded by **10** and **12**, respectively.

*3.3. Ferric Reducing Power Activity*

The data indicated clearly that the thiomethyl parent (**1**) had the superiority as the ferric reducing power agent (Table 3). In a similar manner to the results of DPPH assay, compound **1** recorded the highest ferric reducing power activity of 3567 ± 14.84 µmol.

**Table 2.** Reducing power ability of **1–14**. Data are means $\pm$ standard deviation of triplicate experiments.

| Compound | Reducing Power (Absorbance at 700 nm) |
|----------|----------------------------------------|
| **1** | $1.43 \pm 0.007$ |
| **2** | $0.270 \pm 0.004$ |
| **3** | $0.31 \pm 0.007$ |
| **4** | $0.36 \pm 0.019$ |
| **5** | $0.37 \pm 0.019$ |
| **6** | $0.77 \pm 0.014$ |
| **7** | $0.16 \pm 0.012$ |
| **8** | $0.34 \pm 0.020$ |
| **9** | $0.58 \pm 0.013$ |
| **10** | $0.272 \pm 0.006$ |
| **11** | $0.21 \pm 0.006$ |
| **12** | $0.310 \pm 0.010$ |
| **13** | $0.37 \pm 0.006$ |
| **14** | $0.41 \pm 0.008$ |
| **BHA** | $1.76 \pm 0.062$ |

**Table 3.** Ferric reducing power activity of **1–14**. Data are means $\pm$ standard deviation of triplicate experiments.

| Compound | FRAP Assay (µmol Trolox/100 g) |
|----------|---------------------------------|
| **1** | $3567 \pm 14.84$ |
| **2** | $953 \pm 12.50$ |
| **3** | $882 \pm 7.09$ |
| **4** | $462 \pm 18.18$ |
| **5** | $1482 \pm 8.19$ |
| **6** | $681 \pm 9.07$ |
| **7** | $385 \pm 6.56$ |
| **8** | $315 \pm 7.37$ |
| **9** | $1474 \pm 11.06$ |
| **10** | $542 \pm 10.60$ |
| **11** | $2093 \pm 6.08$ |
| **12** | $564 \pm 12.86$ |
| **13** | $520 \pm 11.50$ |
| **14** | $367 \pm 9.71$ |

Trolox/100 g, while phenoxy and methylsulfonyl parents (**2** and **3**) showed lower ferric reducing capacity of $953 \pm 12.50$ and $882 \pm 7.09$ µmol Trolox/100 g, respectively. Substitution on the structure of **1** led to a large decrease in the reducing activity. In particular, **8** showed the lowest activity of $315 \pm 7.37$ µmol Trolox/100 g, while **6** and **7** exhibited reducing capacity of $681 \pm 9.07$ and $385 \pm 6.56$ µmol Trolox/100 g at the same concentration. However, compounds **5** and **9** displayed good activity ($1482 \pm 8.19$ and $1474 \pm 11.06$ µmol Trolox/100 g) in comparison to **1**. In case of parent **2**, the substitution on its structure with 2-hydroxy-5-methoxy (**11**) resulted in a clear increase in the reducing ability to reach about three-folds ($2093 \pm 6.08$ µmol Trolox/100 g), while with 2-hydroxy-5-methyl and 2-hydroxy-5-nitro groups (**10** and **12**) delivered $542 \pm 10.60$ and $564 \pm 12.86$ µmol Trolox/100 g. Generally, compounds **1**, **5**, **9**, and **2** appeared more active than the other investigated compounds.

### 3.4. ABTS Radical Scavenging Activity

Figure 3 illustrated the ABTS radical scavenging activity of **1–14** in comparison with BHA. The findings presented in Figure 3 confirmed the DPPH scavenging effects presented in Figure 2 and demonstrated the ABTS radical scavenging effects of the parent compounds **1–3** ($88.59 \pm 0.13$, $62.00 \pm 0.24$, and $29.98 \pm 0.13\%$, respectively) and the IC$_{50}$ values of **1** and **2** were 56.44 and 54.34 µg/mL, respectively. It was observed that compound **1** exhibited the highest ABTS radical scavenging activity ($88.59 \pm 0.13\%$) compared to standard BHA

(96.18 ± 0.33%) at the same concentration. Chemical substitution on the structure of **2** with 2-hydroxy-5-methoxy to produce **11** resulted in a slight change in the ABTS radical scavenging activity to reach 53.42 ± 0.14% (IC$_{50}$ = 93.59 µg/mL). Based on all assay findings, the parent methylthio (**1**) demonstrated a significant antioxidant activity comparable to that of BHA. From the electronic chemistry point of view, the excellent activity of **1** relative to the other two parents (**2** and **3**) could be attributed to the electronic enriching of the parent structure and its pi-electron conjugation by the +R effect of the pair of lone-pair of electrons on *S*-atom of –SCH$_3$ group that is reinforced by the hyper conjugation effect of CH$_3$-hydrogen atoms. The contrary was noticed by the drastic lowering in the antioxidant activity of **2** and **3**, where (–R)- and (–I)-effects of the phenoxy or sulfonyl caused the reverse effect occurred by the –SCH$_3$ group in **1**. Further theoretical studies were carried out by using the B3LYP/6-311++g (2d,2p) level of theory to provide us with more insights about molecular antioxidant properties of the targets and to clarify the SAR of compounds **1–14**.

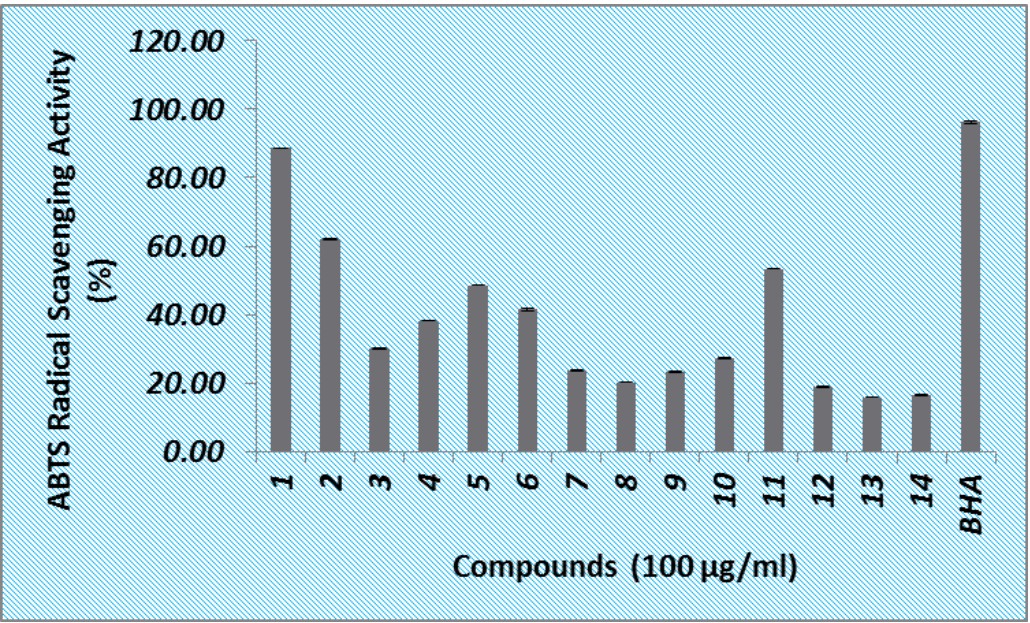

**Figure 3.** ABTS radical scavenging activity of **1–14**. Data are means ± standard deviation of triplicate experiments.

### *3.5. DFT Study*

#### 3.5.1. Geometry Optimization

The ability to describe the scavenging action of antioxidant compounds (**1–14**) requires a detailed understanding of their electrical and structural characteristics. Consequently, various preliminary geometries of examined compounds were selected for further optimization. Using the B3LYP/6-311++g (2d,2p) level of theory and molecular mechanics optimization with MM$^+$ force field delivered in chem 3D, the compounds were geometry optimized. The most stable geometries have been selected for preparation of the radical input geometries and further investigation (Figure S1). Three or fewer initial proposals per compound (**1–14**), have been examined for this purpose.

#### 3.5.2. Global Reactivity Descriptors

Table 4 summarized the reactivity descriptors values, and all reported data are in electron volt (eV). The stability of the molecule is determined by the η value, whilst the *S* parameter provides information about chemical reactivity of substance. Molecules **4–12** have lower estimated η than other examined molecules in the gas phase, indicating their low stabilities. Compounds **4–12** have a greater *S* parameter of 0.025–0.028, indicating their higher charge-transfer mechanism than that of the other substances. The χ was studied for

a better understanding of the charge-transfer reaction. The molecules abilities to attract electrons are represented by their negative of μ. Accordingly, compounds **1**, **2**, **13**, and **14** have low electronegativity, indicating that they are better at giving electrons rather than grabbing them and thus showing low antioxidant activity. The chemical reactivity characteristics of **3**–**12** showed that they are more suitable targets for electron-scavenging processes. It is clear that all targets are highly polarized, demonstrating that the polarity of the environment will have a significant impact on the scavenging electron response. The values of dipole moments are listed in Table 5.

**Table 4.** Global reactivity descriptors at the B3lyp/6-311++g (2d,2p) level of theory.

| Comp. | εHOMO | εLUMO | Gap (kcal) | IP | EA | χ | μ | η | S | ω |
|---|---|---|---|---|---|---|---|---|---|---|
| 1 | −137.818 | −48.098 | 89.72 | 137.818 | 48.098 | 92.958 | −92.958 | 44.86 | 0.022 | 193,821.2 |
| 2 | −146.534 | −39.087 | 107.447 | 146.534 | 39.087 | 92.81 | −92.81 | 53.723 | 0.019 | 231,380.5 |
| 3 | −165.61 | −56.218 | 109.392 | 165.61 | 56.218 | 110.914 | −110.914 | 54.696 | 0.018 | 336,431.6 |
| 4 | −136.092 | −60.491 | 75.601 | 136.092 | 60.491 | 98.292 | −98.292 | 37.801 | 0.026 | 182,600.3 |
| 5 | −136.356 | −61.928 | 74.428 | 136.356 | 61.928 | 99.142 | −99.142 | 37.214 | 0.027 | 182,889.7 |
| 6 | −141.953 | −71.378 | 70.575 | 141.953 | 71.378 | 106.666 | −106.666 | 35.287 | 0.028 | 200,742.4 |
| 7 | −138.991 | −65.781 | 73.21 | 138.991 | 65.781 | 102.386 | −102.386 | 36.605 | 0.027 | 191,864.4 |
| 8 | −137.052 | −59.638 | 77.415 | 137.052 | 59.638 | 98.345 | −98.345 | 38.707 | 0.026 | 187,183.4 |
| 9 | −136.632 | −59.374 | 77.258 | 136.632 | 59.374 | 98.003 | −98.003 | 38.629 | 0.026 | 185,507.1 |
| 10 | −142.229 | −61.162 | 81.067 | 142.229 | 61.162 | 101.696 | −101.696 | 40.533 | 0.025 | 209,598.7 |
| 11 | −137.353 | −56.488 | 80.866 | 137.353 | 56.488 | 96.921 | −96.921 | 40.433 | 0.025 | 189,905.3 |
| 12 | −144.789 | −67.927 | 76.862 | 144.789 | 67.927 | 106.358 | −106.358 | 38.431 | 0.026 | 217,368 |
| 13 | −136.977 | −47.326 | 89.651 | 136.977 | 47.326 | 92.152 | −92.152 | 44.825 | 0.022 | 190,326.7 |
| 14 | −142.957 | −38.052 | 104.905 | 142.957 | 38.052 | 90.504 | −90.504 | 52.453 | 0.019 | 214,821 |
| BHA | −129.76 | −8.555 | 121.206 | 5.627 | 8.555 | 69.159 | −69.159 | 60.603 | 8.786 | 39.457 |

**Table 5.** The O–H or N–H BDE, AIP, PDE, PA, and ETE at the B3LYP/6-311++g (2d,2p) level of theory in the gas phase.

| Comp. | BDE | AIP | PDE | PA | ETE | Dipole Moment | Polarizability (α) |
|---|---|---|---|---|---|---|---|
| 1 | 82.94 | 170.52 | 228.33 | 332.34 | 66.52 | 2.95 | 189.69 |
| 2 | 89.21 | 168.78 | 236.35 | 343.04 | 62.1 | 4.65 | 207.84 |
| 3 | 92.67 | 194.35 | 214.24 | 330.71 | 77.88 | 8.19 | 189.52 |
| 4 | 81.21 | 164.38 | 232.74 | 319.59 | 77.54 | 8.3 | 319.35 |
| 5 | 80.32 | 164.61 | 231.63 | 332.77 | 63.46 | 9.37 | 311.64 |
| 6 | 82.7 | 172.24 | 226.38 | 305.49 | 93.13 | 8.72 | 324.28 |
| 7 | 81.14 | 167.79 | 229.27 | 314.04 | 83.02 | 7.85 | 326.89 |
| 8 | 81.51 | 165.05 | 232.38 | 316.55 | 80.87 | 7.9 | 310.27 |
| 9 | 75.61 | 164.04 | 227.48 | 313.97 | 77.56 | 8.49 | 329 |
| 10 | 83.1 | 195.28 | 203.74 | 340.33 | 58.69 | 8.16 | 359.71 |
| 11 | 77.49 | 164.84 | 228.57 | 325.5 | 67.91 | 7.26 | 335 |
| 12 | 84.53 | 171.25 | 229.19 | 308.69 | 91.75 | 11.21 | 439.88 |
| 13 | 81.7 | 171.87 | 225.74 | 112.7 | 284.92 | 8.84 | 312.56 |
| 14_ArH | 84.57 | 173.92 | 226.57 | 338.74 | 61.74 | 11.91 | 383.06 |
| 14_RH | 95.81 | 173.92 | 237.81 | 338.74 | 72.99 | 11.91 | 383.06 |
| BHA | 82.929 | 132.667 | 266.122 | 309.53 | 88.519 | 3.156 | 172.023 |

### 3.5.3. Frontier Molecular Orbitals

Figure 4 presented the distribution of HOMO and LUMO for the studied compounds in the gas phase. The HOMO plots revealed that HOMOs are concentrated on the triazole ring and distributed on –NH and -S- for molecules **1**–**14** as well. Thus, free radicals are more likely to attack this -S- group, losing an electron in the process. On the other hand, LUMOs distribution demonstrated that the carbons of the phenol and benzoic acid rings

heavily contributed to this instance, and no contributions appeared on –OH, –NH, or -S-. These areas may be useful for nucleophiles molecular reactions.

Molecules with a lower molecular orbital gap at the frontier are more polarized. Intermolecular charge transfer among electron donors and electron acceptors can occur in this instance, which may affect the molecule's biological activity [49]. The data in Table 4 revealed that molecules **4, 5, 6** and **7** have a low HOMO–LUMO gap, indicating their high degree of polarizability. Furthermore, these values are identical for molecules **4, 5, 6** and **7**, implying that they have a similar biological activity.

### 3.5.4. Molecular Electrostatic Potential (MEP)

MEP is a vital descriptor to confirm the related parts of the molecular reactivity system, and it can be calculated using several methods. The three-dimensional MEP surface provides us with more information, such as location, shape, size of the positive, negative, and neutral electrostatic potentials, which help to understand the physicochemical properties and the relationship of molecular structure reactivity towards electrophilic and nucleophilic attacks. The highest negative electronic potential, indicated by red in Figure 5, is the favored site for the electrophilic attack. Meanwhile, the positive electrostatic potential, shown in blue (Figure 5), will be attracted by radicals. As illustrated in Figure 5, strong electrostatic potential zones can be found at the oxygen of hydroxyl groups, oxygen of carboxylic acid groups, and sulfur in thiol groups, whereas low electropositive potential may be found mostly on H of hydroxyl and amine groups. The values of blue color codes in compounds **1, 2, 3, 13,** and **14** ranged from $5.85 \times 10^{-2}$ to $1.31 \times 10^{-1}$, indicating that H in **13** is the preferred location for nucleophilic interactions. However, the codes for H in **9, 5,** and **7** are $+7.831 \times 10^{-2}$, $+7.626 \times 10^{-2}$ and $+8.151 \times 10^{-2}$, showing its preferential for nucleophilic interacted. The electrostatic potential intensity region around the oxygen of the ring makes it a suitable radical trap.

| Comp. | HOMO | LUMO |
|:-----:|:----:|:----:|

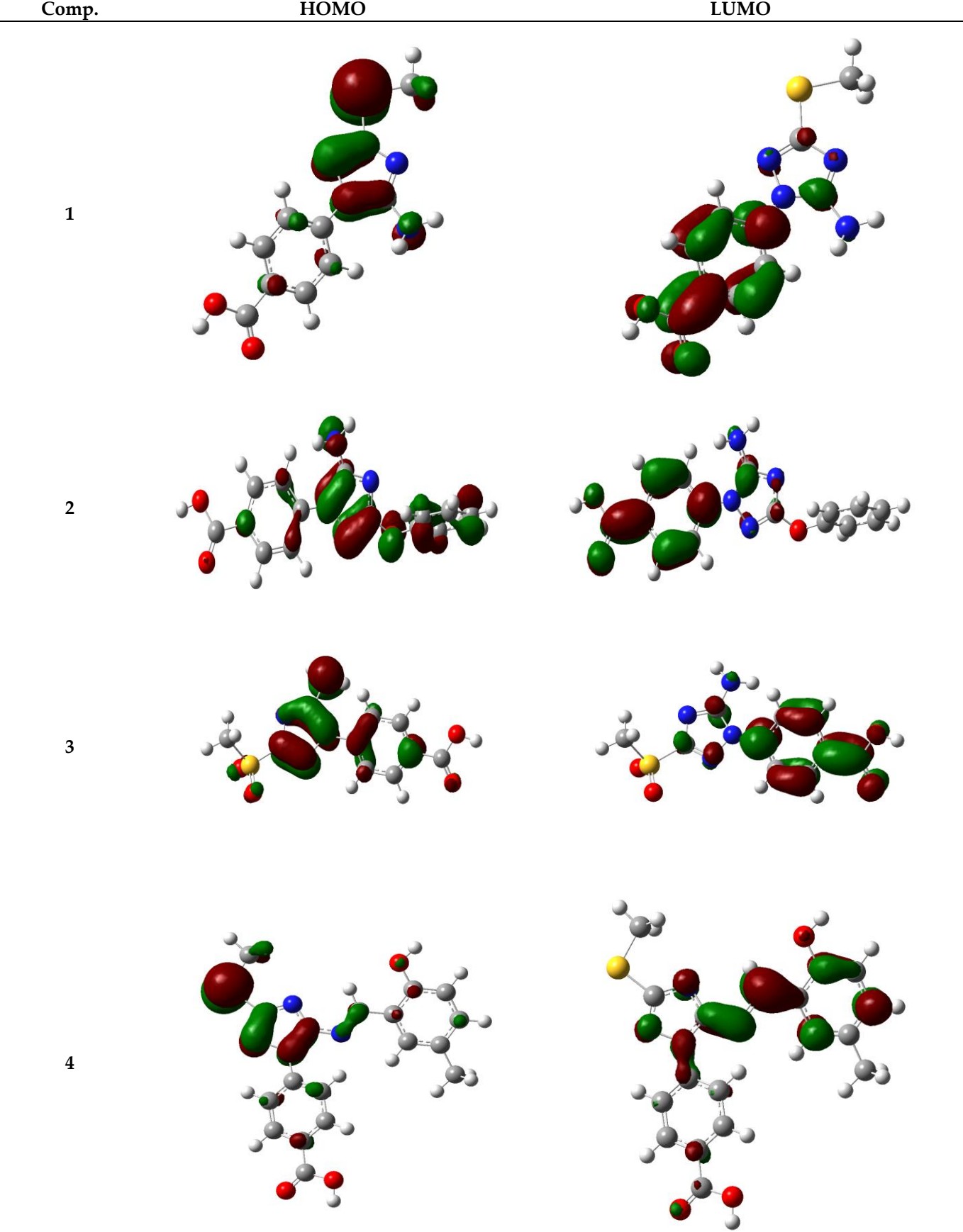

**Figure 4.** *Cont.*

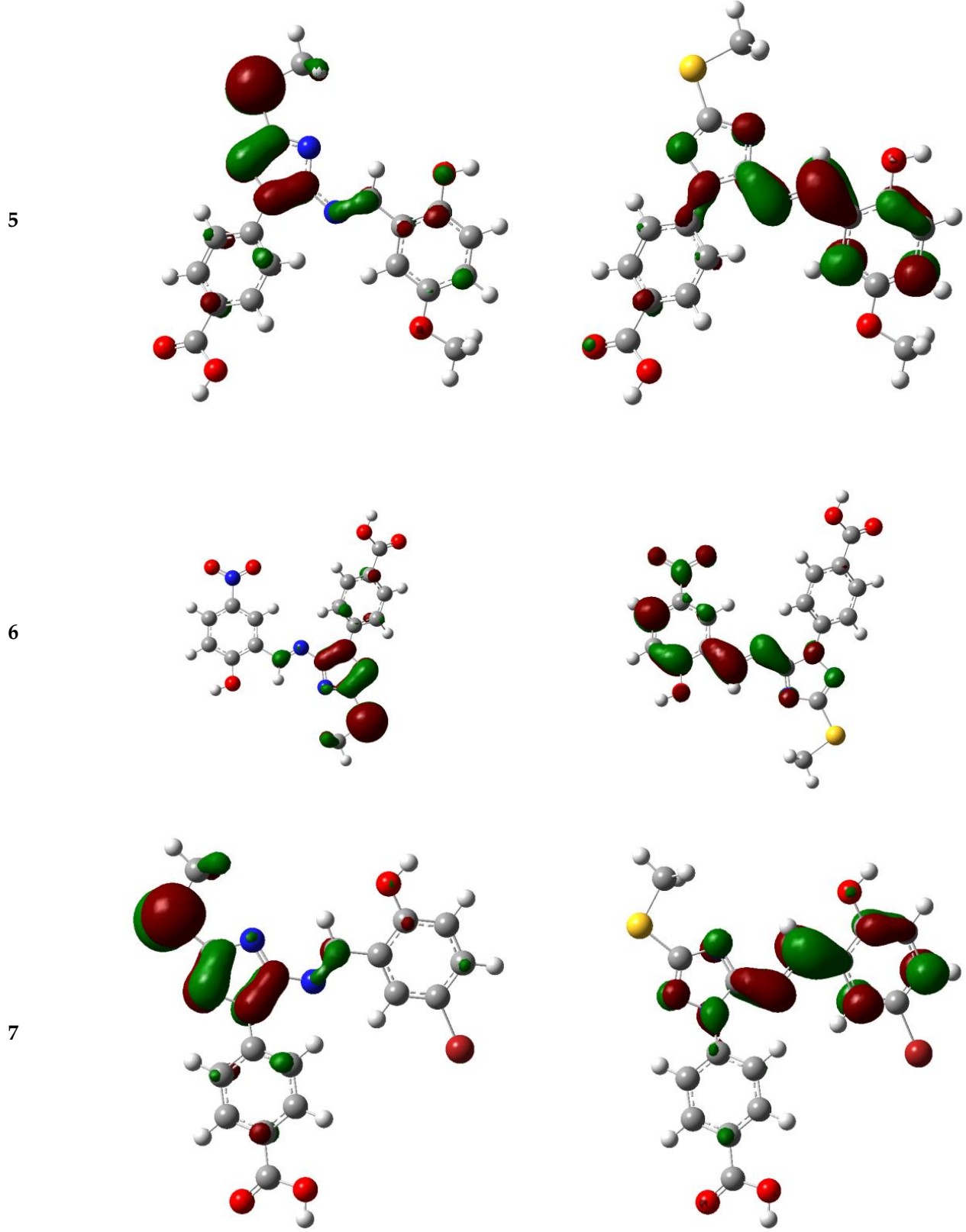

**Figure 4.** *Cont.*

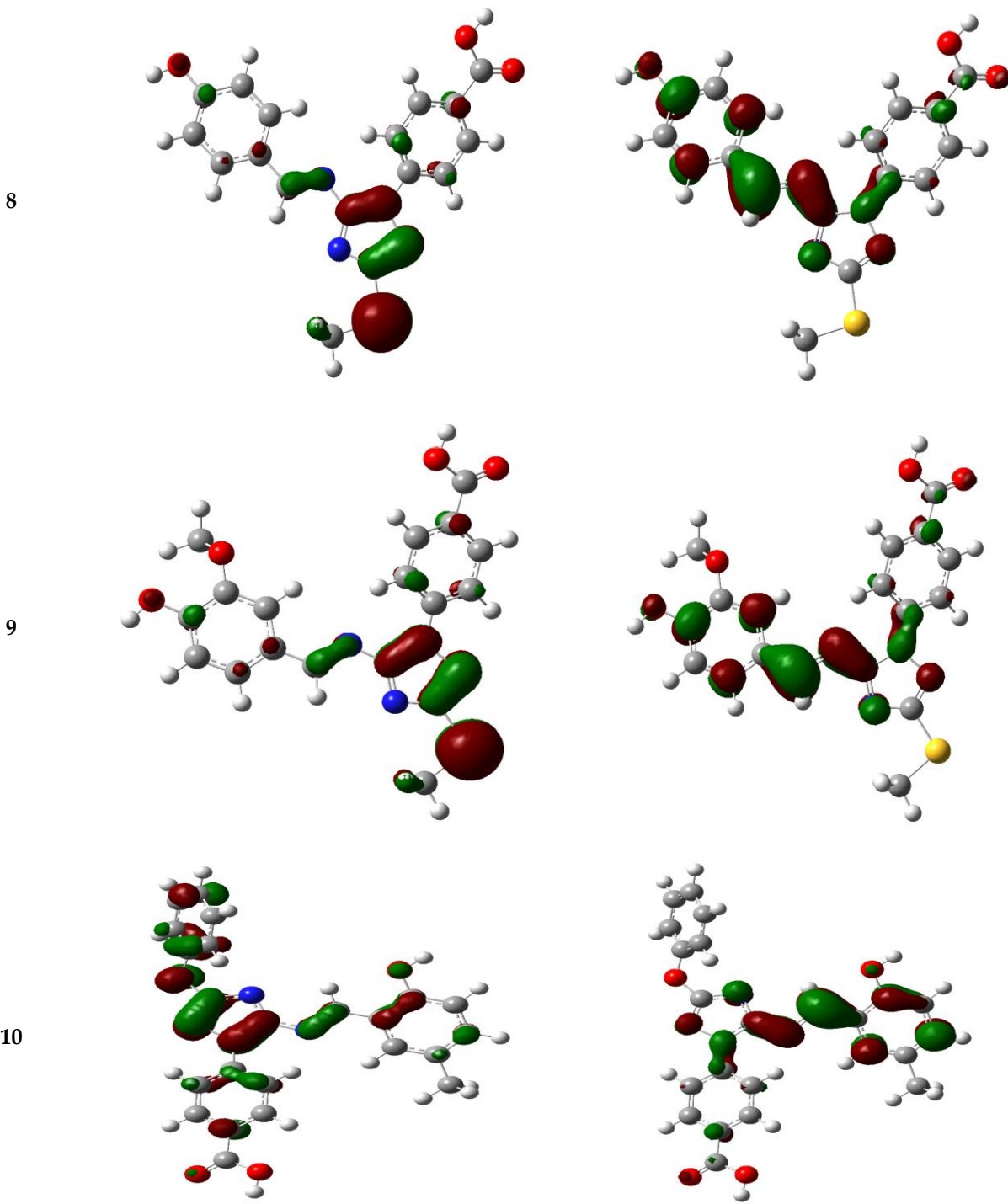

**Figure 4.** *Cont.*

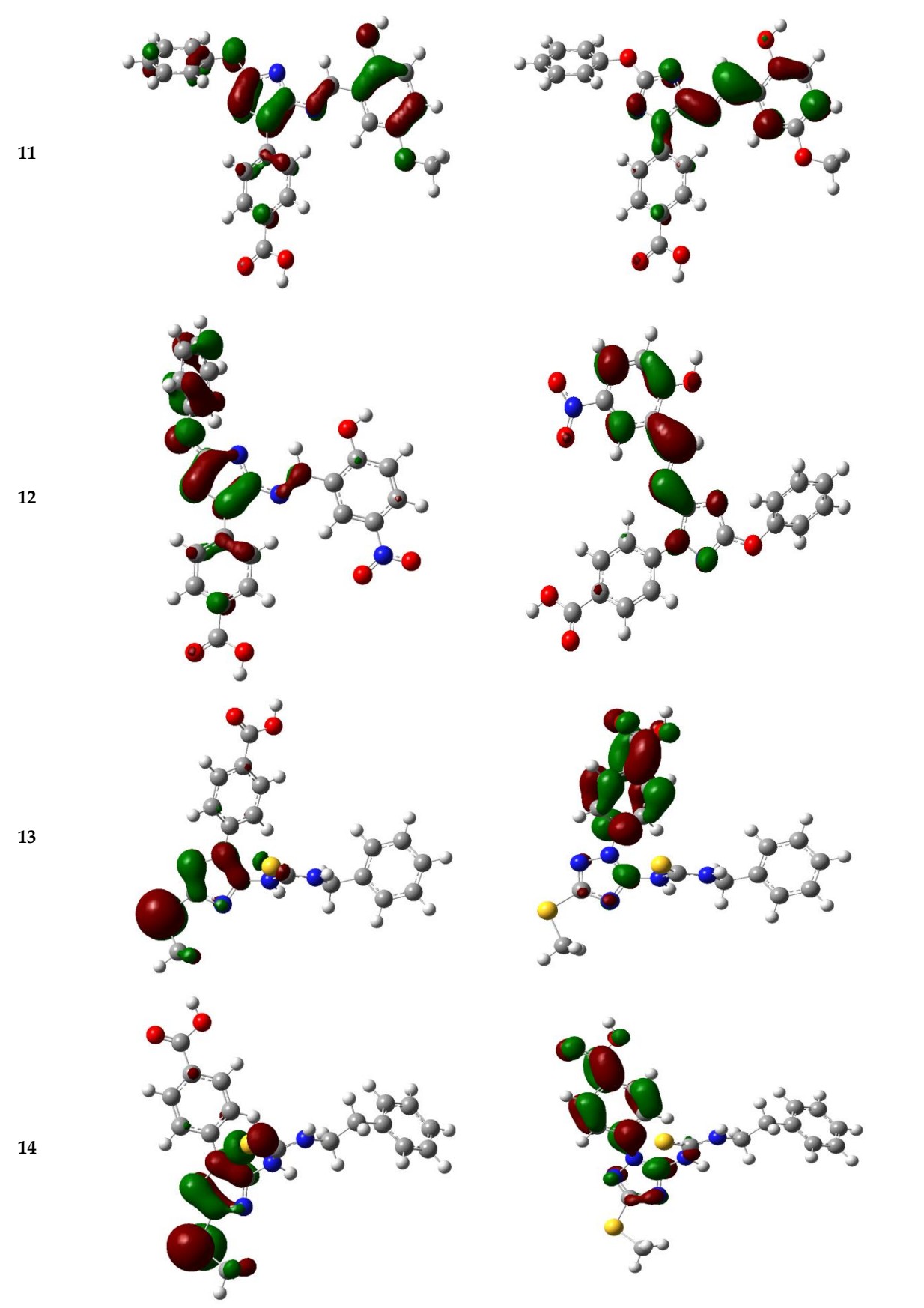

**Figure 4.** HOMO and LUMO plots for **1**−**14** in the gas phase.

| Comp. | Electrostatic Potential | | Electrostatic Potential |
|---|---|---|---|
| | | 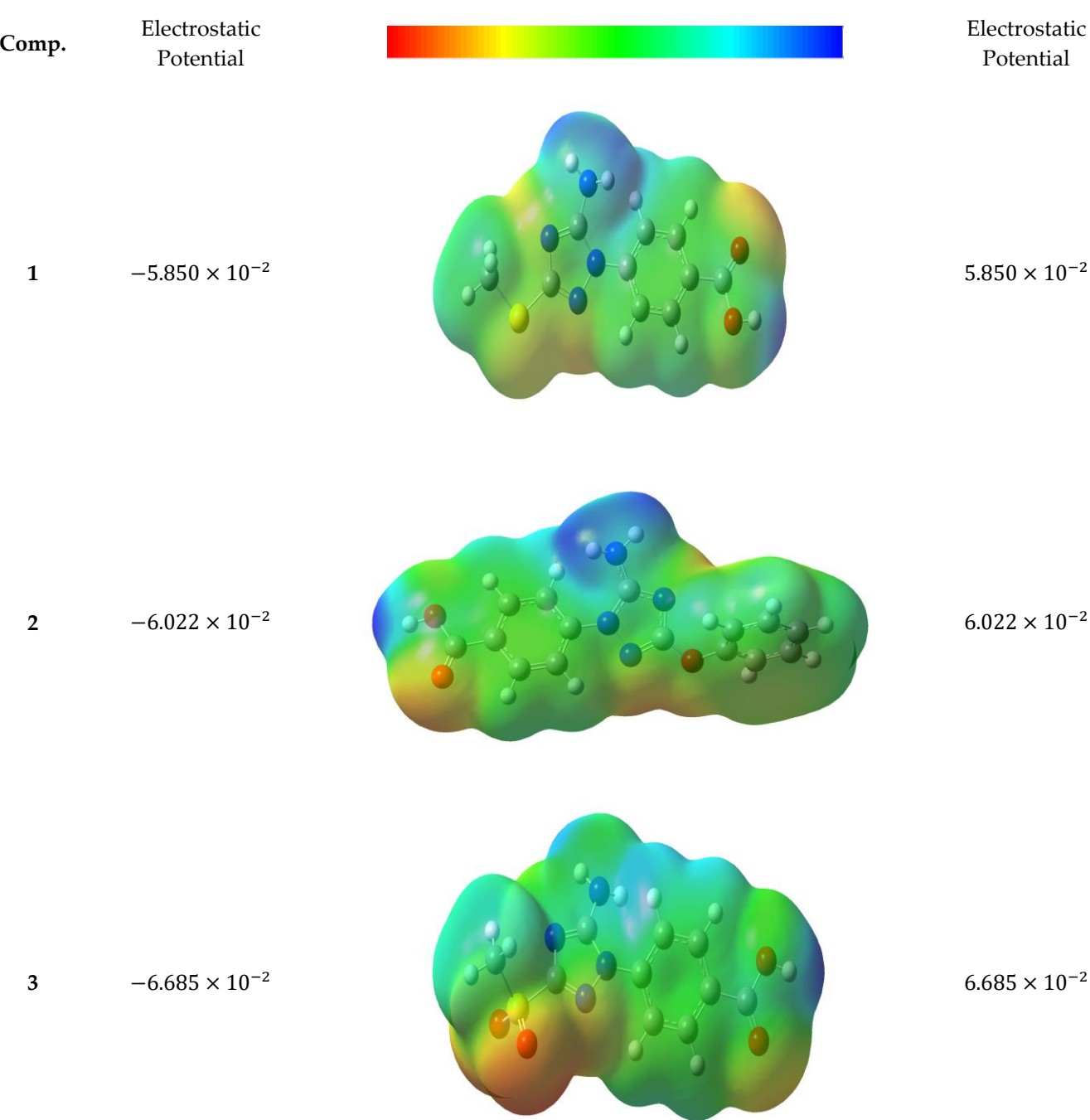 | |
| **1** | $-5.850 \times 10^{-2}$ | | $5.850 \times 10^{-2}$ |
| **2** | $-6.022 \times 10^{-2}$ | | $6.022 \times 10^{-2}$ |
| **3** | $-6.685 \times 10^{-2}$ | | $6.685 \times 10^{-2}$ |

**Figure 5.** *Cont.*

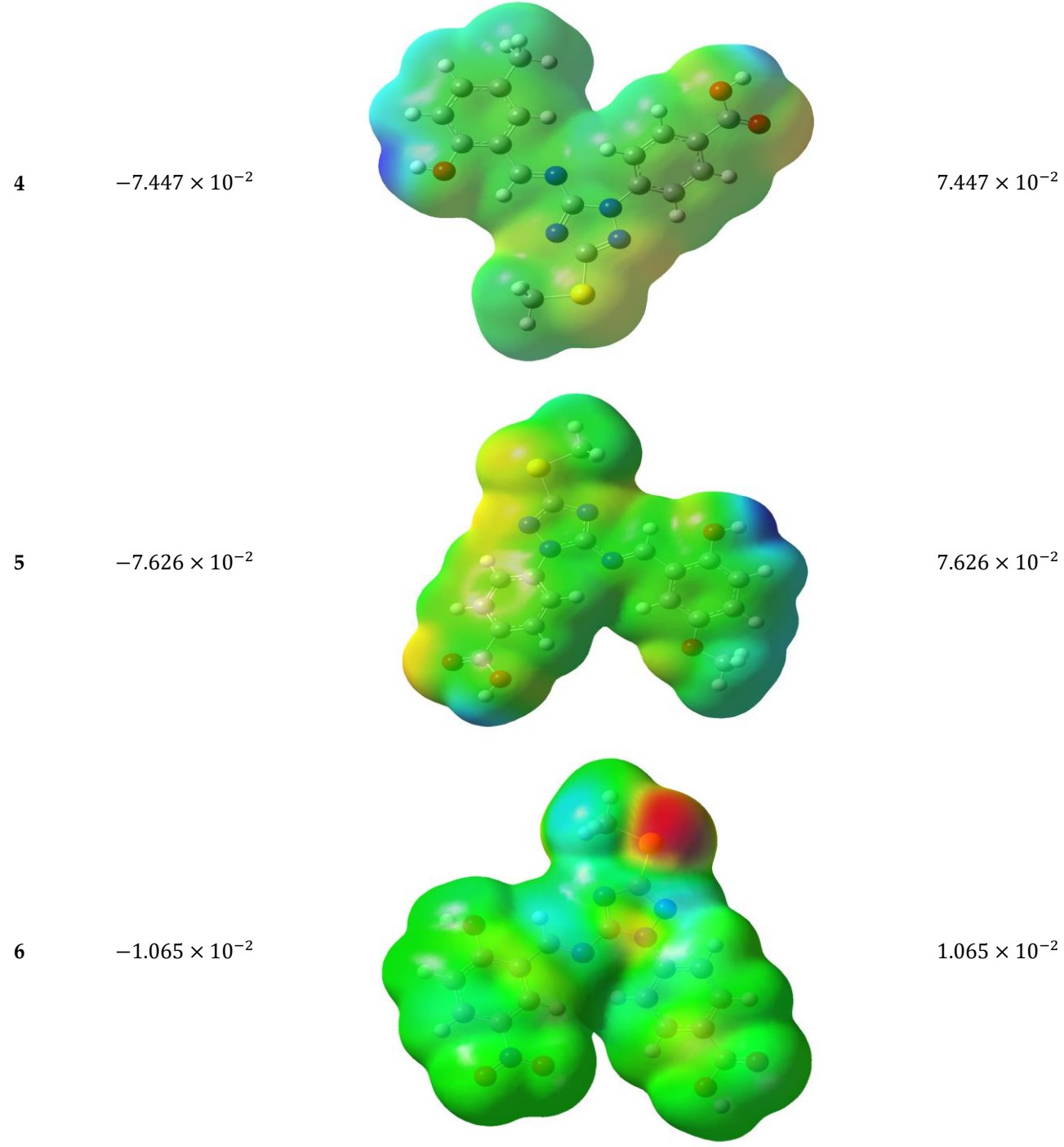

**4**  $-7.447 \times 10^{-2}$    $7.447 \times 10^{-2}$

**5**  $-7.626 \times 10^{-2}$    $7.626 \times 10^{-2}$

**6**  $-1.065 \times 10^{-2}$    $1.065 \times 10^{-2}$

**Figure 5.** *Cont.*

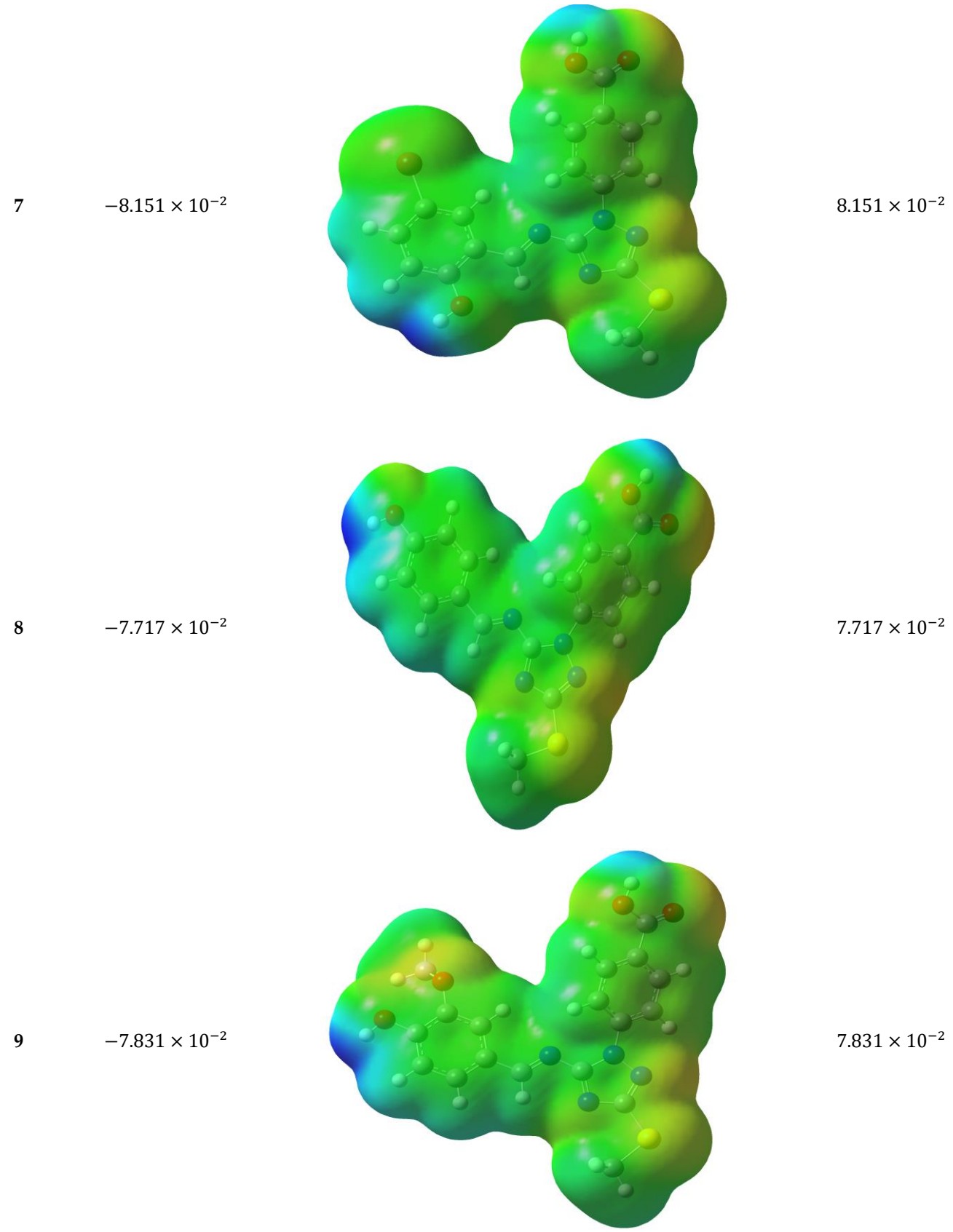

**7**    $-8.151 \times 10^{-2}$    $8.151 \times 10^{-2}$

**8**    $-7.717 \times 10^{-2}$    $7.717 \times 10^{-2}$

**9**    $-7.831 \times 10^{-2}$    $7.831 \times 10^{-2}$

**Figure 5.** *Cont.*

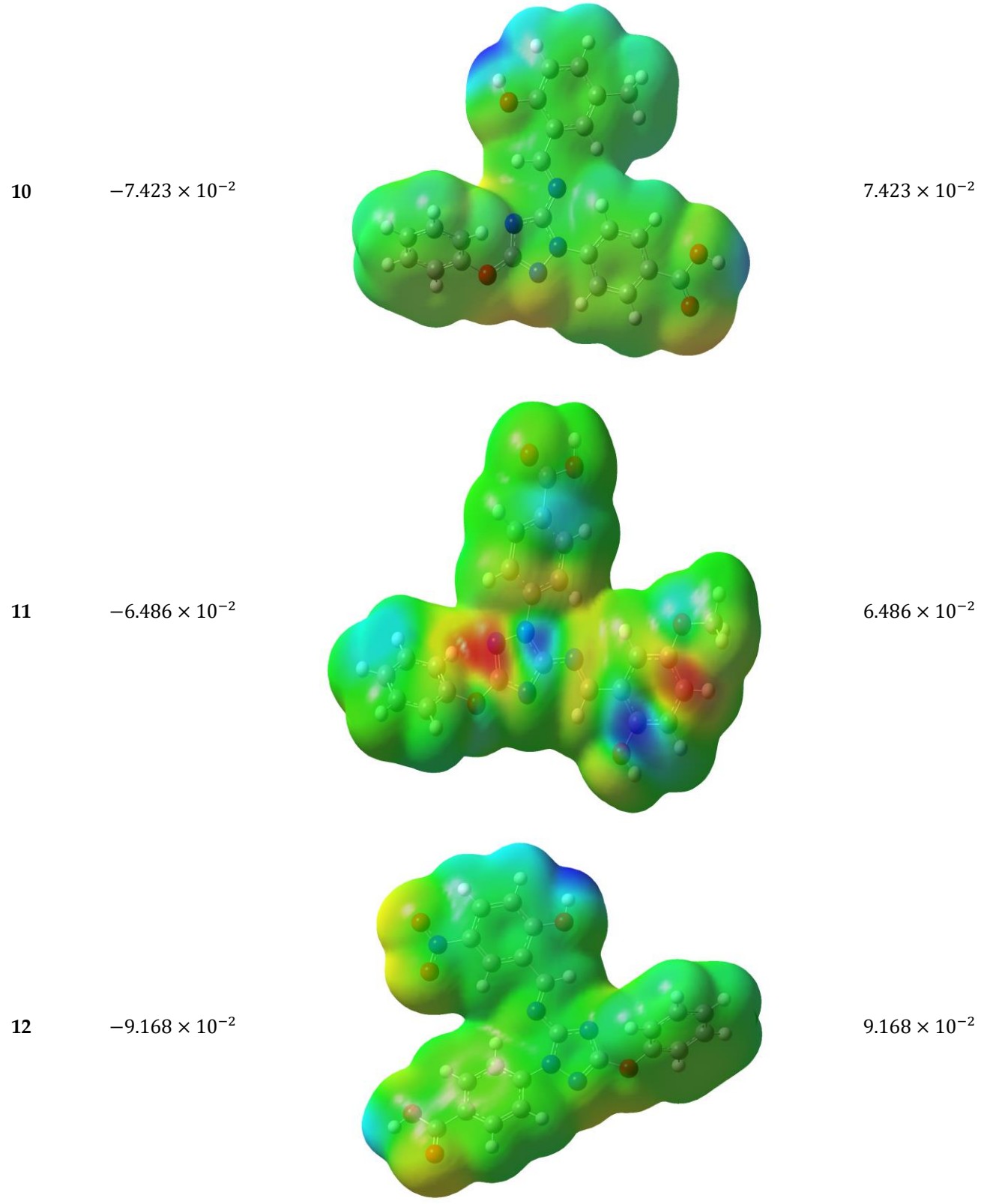

**10**  $-7.423 \times 10^{-2}$  $7.423 \times 10^{-2}$

**11**  $-6.486 \times 10^{-2}$  $6.486 \times 10^{-2}$

**12**  $-9.168 \times 10^{-2}$  $9.168 \times 10^{-2}$

**Figure 5.** *Cont.*

**13**          $-1.31 \times 10^{-1}$

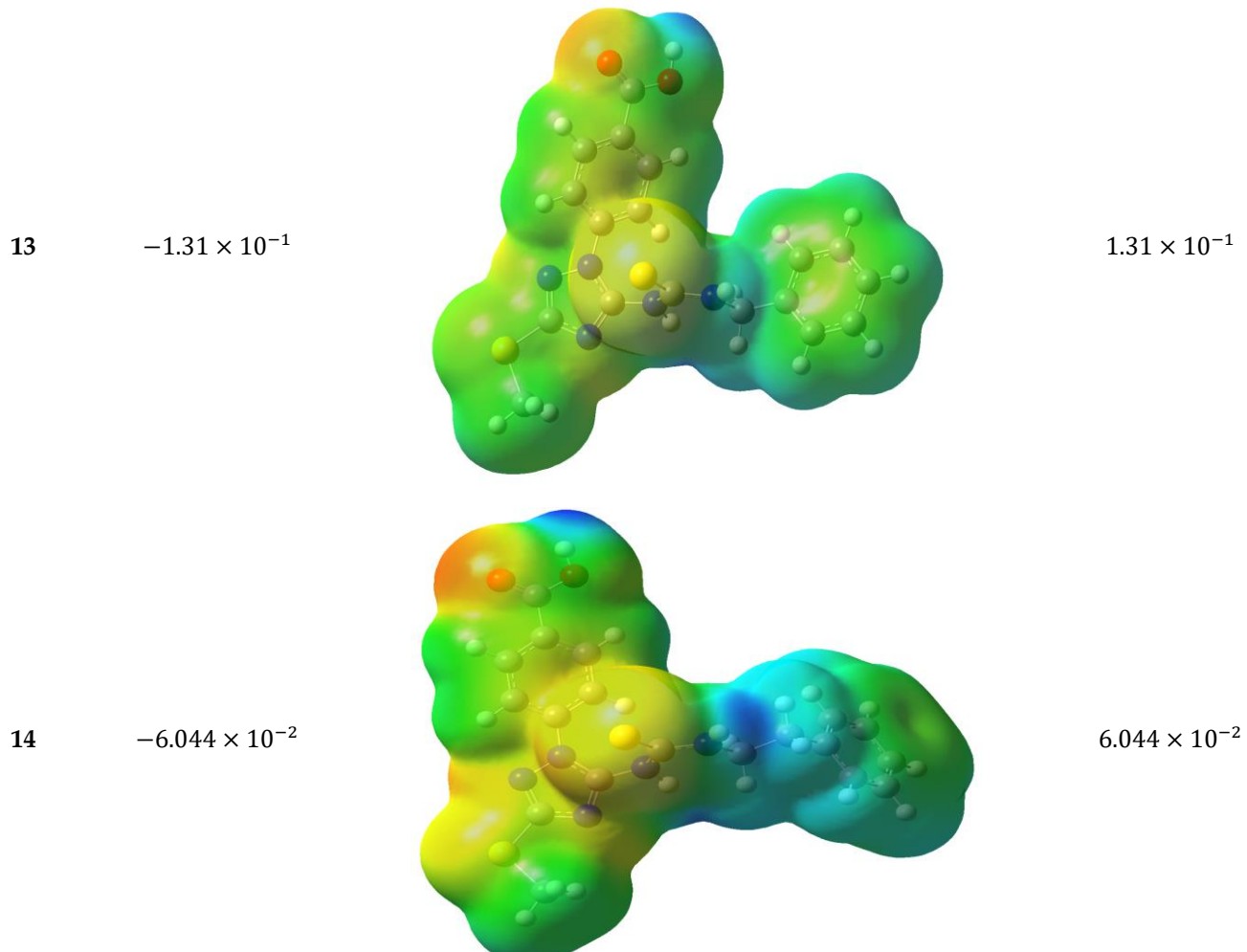

$1.31 \times 10^{-1}$

**14**          $-6.044 \times 10^{-2}$

$6.044 \times 10^{-2}$

**Figure 5.** Molecular electrostatic potential of **1**–**14**.

3.5.5. Antioxidant Mechanisms of the Target Molecules **1**–**14**

It is well established that the HAT mechanism is determined by the BDE values, which indicate the capacity of an XH (X=C, O, or N) moiety to donate its hydrogen atom and so create a radical. Increasing antioxidant potency is associated with decreased BDE of the relevant X-H bond [50]. Thus, in this part, the BDE values of the weakest X-H bonds of each molecule were calculated and presented in Table 5.

Obviously, based on the data presented in Table 5, BDE (X-H)s where X denotes the elements C, O, or N normally ranged between 75.61 and 95.81 kcal.mol$^{-1}$. The phenolic methoxy group in **5**, **9**, and **11** in the *ortho* positions may have played a key role in the antioxidant activity. Therefore, the O–H (phenolic) bonds have the lowest BDE values, ranging from 75.61 to 80.319 kcal/mol, indicating their low stability. Compounds **4** and **10** are characterized by the phenolic alkyl group, which played as an activator for the bond dissolution process and enhanced positively in the antioxidant properties. This is in agreement with the in vitro study of antioxidants. In contrast, the presence of nitro and bromo substituents in **6**, **12**, and **7** influenced negatively on the activity (acting as a deactivator for bond dissolution) resulting in decreasing the antioxidant activity. Thus, the O–H (phenolic) bonds with higher BDE are considered more stable. Based on the gas phase BDE values, the H-donation abilities of the targets were ordered as **14_RH > 3 > 2 > 12 ≈ 14_ArH > 10 > 1 ≈ 6 > 13 ≈ 8 ≈ 4 ≈ 7 > 5 > 11 > 9**. The solvent effect tests were performed on these compounds using methanol and water. Compounds **5**, **9**, and **11** have the lowest BDE values for O–H at 80.322, 75.61 and 77.49 kcal.mol$^{-1}$, respectively (Table 5). Solvation effect assessments were performed on these compounds

using methanol and water. Shifting BDE values obviously demonstrated that the polarity of the solvents affects the hydrogen-donating ability. Thus, methanol and water solvents were utilized in this investigation as the environment in which BDEs were calculated. The antioxidant activity and radical trapping abilities were performed in methanol [51,52]. The obtained findings of **5**, **9**, and **11** showed a decrease in BDEs (X–H, X = O, N) values in the solvents in the range of 4.2 to 11.4 kcal.mol$^{-1}$ when the compounds are exposed to methanol solvent. The O–H bonds are still the easiest dissociation links in the selected **5**, **9**, and **11** (Table 6).

**Table 6.** The O–H BDE, IP, PDE, PA, and ETE at the B3LYP/6-311++g (2d,2p) level of theory in methanol solvent.

| Comp. | BDE | IP | PDE | PA | ETE |
|---|---|---|---|---|---|
| 5 | 78.8146275 | 169.60321 | 225.12943 | 300.0272025 | 94.7054375 |
| 9 | 78.2611725 | 135.4063425 | 258.7728425 | 283.7410675 | 110.4381175 |
| 11 | 83.79384 | 136.969445 | 262.7424075 | 297.975905 | 101.7359475 |

As stated in the SETPT mechanism, the ionization of an antioxidant molecule is the initial step, and hence the term "AIP" is used to determine the antioxidant's ability (acting as an electron donor). The higher the antioxidant activity, the easier it is to move electrons, and the lower the AIP value (Table 5). The sequences of vertical AIP values in gas phase were ordered as **9 ≈ 4 ≈ 5 ≈ 11 < 8 < 7< 2 < 1 < 12 ≈ 13 < 6 < 14_ArH = 14_RH < 3 <10** (Table 5).

The loss of a proton from the cation radical created in the first stage is the final step in the SETPT mechanism. The PDE parameter was calculated to determine the thermodynamically favorable X–H (X=O, and N) group for deprotonation. Table 5 presented the calculated results.

The easiest deprotonation of the X–H (X=O, and N) bond is usually found in compounds having the lowest PDE values. In the investigated compounds, the lowest PDE values are found in **10**, **3**, **13**, and **6** (203.74, 214.244, 225.74, and 226.378 kcal.mol$^{-1}$, respectively). Moreover, the antioxidant potential is determined by the sum of PDE and IP [53]. Based on the calculated values, compounds **5**, **9**, and **11** are found to have good antioxidant properties, however, **9** appeared to have the lowest PDE + AIP value of 391.52 kcal.mol$^{-1}$ in accordance with Table 5.

As a result, regardless of whether the reaction followed the HAT or SETPT mechanism, this molecule might be the superior antioxidant. Comparison with the calculated BDE and AIP values of **9** (75.61 and 164.045 kcal.mol$^{-1}$ at B3LYP/6-311++G (d,p), respectively) was made, and concluded that **9** slightly matched with in vitro biological experiments (FRAP and reducing power ability).

PA and ETE are essential parameters to describe the SPLET mechanism. A lower PA value indicates stronger antioxidant capacity. The PA values of the compounds were initially analyzed by the B3LYP/6-311$^{++}$G (d,p), as stated above (Table 5). The PA values of O–H bonds are lower than those of N-H bonds, as seen in Table 5 The lowest PA values for O–H bonds in the gas phase in **6** and **9** are 305.49 and 313.97 kcal.mol$^{-1}$, respectively. While for N–H bonds in the gas phase, the lowest PA values in **2** and **3** are 343.04 and 330.71 kcal.mol$^{-1}$, respectively (Table 5).

The final step in the SPLET mechanism is determined by the ETE. Table 5 showed that the PAs are significantly greater than the ETE values of the gas. The single electron transfer process from the neutral form is therefore less favorable than the procedure from the anionic form in this case. This conclusion is consistent with reported studies [54,55]. In fact, ETE values in the examined solvent (methanol) are significantly higher than those found in the gas phase. In all of the examined environments, **9** showed significant antioxidant property based on the computed PA and ETE values of both steps in the SPLET mechanism, with a total PA + ETE value of roughly 391.53 kcal.mol$^{-1}$. This is consistent with our findings in the HAT and SETPT mechanisms.

## 4. Conclusions

The antioxidant activity of the targets **1**–**14** was successfully evaluated by DPPH, reducing power capability, FRAP, and ABTS assays. The obtained results revealed that compounds **1**–**14** exhibited scavenging activity ranging from low to high; however, **1** showed the best antioxidant results experimentally. DFT studies were performed with the basis set B3LYP/6-311++g (2d,2p) level of theory, to understand the antioxidant activity of **1**–**14**. Based on the computed PA and ETE values of both steps in SPLET mechanism, compound **9** appeared to have good antioxidant activity with a total PA + ETE value of roughly 391.53 kcal.mol$^{-1}$ and was consistent with our findings in the HAT and SETPT mechanisms; thus, it appeared to be covenant with experimental results. The calculated HOMO-LUMO gaps and different antioxidant descriptors for **1**–**14** were participated to clarify their antioxidant properties.

**Supplementary Materials:** The following are available online at https://www.mdpi.com/article/10.3390/app112411642/s1. Figure S1: The full optimized geometries of the compounds **1**–**14**.

**Author Contributions:** Conceptualization, R.A.-S and H.A.A.; methodology, H.A.A.T., R.A.-S. and A.H.B.; validation, H.A.A.T., R.A.-S., M.M., A.H.B. and H.A.A.; investigation, H.A.A.T.; resources, R.A.-S., H.A.A. and M.M.A.; writing—original draft preparation, R.A.-S. and H.A.A.; writing—review and editing, R.A.-S., H.A.A., H.A.A.T., M.M. and M.M.A.; supervision, R.A.-S. All authors have read and agreed to the published version of the manuscript.

**Funding:** The authors extend their appreciation to Research Supporting Project, King Saud University, Riyadh, Saudi Arabia, for funding this work through grant no. RSP-2021/353.

**Institutional Review Board Statement:** Not applicable.

**Informed Consent Statement:** Not applicable.

**Data Availability Statement:** The data presented in this study are available on reasonable request from the corresponding author.

**Conflicts of Interest:** The authors declare that they have no known competing financial interests.

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
