# Peer review of "Biological Evaluation of 4-(1H-triazol-1-yl)benzoic Acid Hybrids as Antioxidant Agents: In Vitro Screening and DFT Study"

_applsci, doi:10.3390/app112411642_

Round 1

Reviewer 1 Report

Dear Authors,

I have just finished reconsidering your manuscript and although the reply letter to my comments is missing, I still came to my own conclusions.

My major concern about this study regarded the lack of originality and scientific relevance, since it was (and it is, because nothing has been changed) nothing more than the umpteenth work by Abuelizz and Al-Salahi concerning the evaluation of the antioxidant power of triazole derivatives using the same tests, the same procedures, and the same working scheme. I believed (and my opinion remains the same, because no new information has been added) that this work is poor in scientific contents, I don't think it will attract much attention among researchers. Additionally, except for compound 1 (DPPH) all the other compounds have proven a very limited antioxidant power, thus establishing that also the reported results are almost uninteresting.  Only for these reasons I believe that the present manuscript is not suitable for publication.

In addition, concerning other criticisms, I noted that 

the Authors DID NOT follow my suggestion of not indicating the compounds with specific numbers since the readers cannot match the number to the structure. I suggested to avoid using numbers and simply talking about "14 compounds of type ...", and again, "One of these compounds was more active than ...", and “three of these compounds were more active than the first mentioned… " but authors DID NOT.

The authors DID NOTt create a Table reporting the most important biological active triazoles developed both at study level and as therapeutics, their pharmacological activity, related in vitro/in vivo assay performed, dosages, literature references.

The authors DID NOT insert in the first column of the Table which they did not create the chemical structure of the reported compounds.

The authors DID NOT include in the Introduction a bar graph indicating how the scientist’s interest in the triazole nucleus is increased in the last three decades, in terms of published articles.

4) The authors DID NOT correct the titles of all the sub sections according to the template.

When the authors tried to follow my suggestion they wreaked havoc by further worsening the quality of the manuscript. In fact, when they tried to change the captions of the Figures 2 and 3 to titles of tables, those titles were not inserted correctly. See pages 6 and 7.

The authors DID NOT correct al Fig.  in Figure.

Even if the list of references has been changed it is still not good because it does not fully respect the App. Sci template.

Apart from the lack of originality and lack of relevance of the contents, as well as uninteresting results, the authors didn't even fulfill most of my requests. My opinion remains the same as in the first report.

Author Response

Dear Editor of Applied Sciences

Thank you so much for your fine cooperation.

According to the reviewers 1 and 3  comments, we have corrected the manuscript.

All corrections are indicated by yellow color as following

Reviewer 1

n addition, concerning other criticisms, I noted that 

the Authors DID NOT follow my suggestion of not indicating the compounds with specific numbers since the readers cannot match the number to the structure. I suggested to avoid using numbers and simply talking about "14 compounds of type ...", and again, "One of these compounds was more active than ...", and “three of these compounds were more active than the first mentioned… " but authors DID NOT.

Response: thank you so much for your comments, however,  sometimes it is better to specify the compounds by their numbers in the abstract.

The authors DID NOTt create a Table reporting the most important biological active triazoles developed both at study level and as therapeutics, their pharmacological activity, related in vitro/in vivo assay performed, dosages, literature references.

Response: As you know triazole nucleus is building block for construction of many bioactive compounds with a variety of biological activities. So we added three references [1-3] to indicate the important of this nucleus. And I think  we need more tables to demonstrate the biological activity of the triazole as you request, the added references [1-3] are enough in this concern .  

The authors DID NOT insert in the first column of the Table which they did not create the chemical structure of the reported compounds.

Response, we added table 1 for the chemical structures of the targets 1-14

The authors DID NOT include in the Introduction a bar graph indicating how the scientist’s interest in the triazole nucleus is increased in the last three decades, in terms of published articles.

Response, thanks a lot for your question, we have mentioned the importance of the triazole form the beginning of the introduction till the end as generally.

4) The authors DID NOT correct the titles of all the sub sections according to the template.

Response,  it was already done.

When the authors tried to follow my suggestion they wreaked havoc by further worsening the quality of the manuscript. In fact, when they tried to change the captions of the Figures 2 and 3 to titles of tables, those titles were not inserted correctly. See pages 6 and 7.

 Response: we have corrected them

The authors DID NOT correct al Fig.  in Figure.

Even if the list of references has been changed it is still not good because it does not fully respect the App. Sci template.

Response; All references are corrected according to the journal style.

Reviewer 2 Report

The authors respond to the reviewers' questions and suggestions to the text, improving the quality of the manuscript and the presentation of results.

Author Response

Thank you so much for your reviewing.

Reviewer 3 Report

In the manuscript Salahi and co-worker report the antioxidant activity of 4-(1H-triazol-1-yl)benzoic acid hybrids and DFT calculations. Manuscript is written well, and I recommend the manuscript after the following minor revision. 

  1. In the abstract please abbreviate DPPH, FRAP, ABTS, BHA. 
  2. In scheme 1 arrange the compounds numbering in the proper way it looks scattered all around there. 
  3. In Figure1 mentioned the numbering of reported triazole derivative and mentioned in the text with reference appropriately. 
  4. In the line 114-117 the value of H(e-)vacuum should be consistent after decimal. 
  5. In the entire manuscript Figure or Fig. should be consistence. 
  6. Author should mention IC50 of at least most potent compound in both DPPH and ABTS radical scavenger activity. 
  7. Author should be mentioned in the figure 2 and figure 3 about which cells they have used. 

Author Response

Dear Editor of Applied Sciences

Thank you so much for your fine cooperation.

According to the reviewers 1 and 3  comments, we have corrected the manuscript.

All corrections are indicated by yellow color as following

Reviewer 3

In the manuscript Salahi and co-worker report the antioxidant activity of 4-(1H-triazol-1-yl)benzoic acid hybrids and DFT calculations. Manuscript is written well, and I recommend the manuscript after the following minor revision. 

  1. In the abstract please abbreviate DPPH, FRAP, ABTS, BHA. 

Response, it was done in the abstract and indicated by yellow color

  1. In scheme 1 arrange the compounds numbering in the proper way it looks scattered all around there. 

Response, we have added table 1 to indicate the chemical structures of the targets and adjusted the scheme 1

  1. In Figure1 mentioned the numbering of reported triazole derivative and mentioned in the text with reference appropriately.

Response, we mentioned the numbering of triazole in figure 1 and in the text also.

  1. In the line 114-117 the value of H(e-)vacuum should be consistent after decimal. 

Response: They were corrected.

  1. In the entire manuscript Figure or Fig. should be consistence. 

Response, According to his place in the sentence we added it as figure or Fig.

  1. Author should mention IC50 of at least most potent compound in both DPPH and ABTS radical scavenger activity. 

Response: we have added them in the text.

  1. Author should be mentioned in the figure 2 and figure 3 about which cells they have used

Response: no cells are used.  DPPH and ABTS are chemical compounds to determine the antioxidant activity